# Electronic Structure of Lr$^+$ (Z = 103) from Ab Initio Calculations

Harry Ramanantoanina [1,2,*], Anastasia Borschevsky [3], Michael Block [1,2,4] and Mustapha Laatiaoui [1,2]

1   Department Chemie, Johannes Gutenberg-Universität, Fritz-Strassmann Weg 2, 55128 Mainz, Germany; m.block@gsi.de (M.B.); mlaatiao@uni-mainz.de (M.L.)
2   Helmholtz-Institut Mainz, Staudingerweg 18, 55128 Mainz, Germany
3   Van Swinderen Institute for Particle Physics and Gravity, University of Groningen, Nijenborgh 4, 9747 Groningen, The Netherlands; a.borschevsky@rug.nl
4   GSI Helmholtzzentrum für Schwerionenforschung, Planckstrasse 1, 64291 Darmstadt, Germany
*   Correspondence: haramana@uni-mainz.de

**Abstract:** The four-component relativistic Dirac–Coulomb Hamiltonian and the multireference configuration interaction (MRCI) model were used to provide the reliable energy levels and spectroscopic properties of the Lr$^+$ ion and the Lu$^+$ homolog. The energy spectrum of Lr$^+$ is very similar to that of the Lu$^+$ homolog, with the multiplet manifold of the $7s^2$, $6d^17s^1$ and $7s^17p^1$ configurations as the ground and low-lying excited states. The results are discussed in light of earlier findings utilizing different theoretical models. Overall, the MRCI model can reliably predict the energy levels and properties and bring new insight into experiments with superheavy ions.

**Keywords:** MRCI; electronic structure; electric dipole transitions

## 1. Introduction

A new development in the field of atomic spectroscopy and ion mobility has been recently proposed under the name of Laser Resonance Chromatography (LRC) [1], a method that gained interest in particular because of its potential applicability to superheavy elements. In this method, optical resonances are identified based on resonant optical pumping of ions drifting in diluted helium [1,2]. The optical pumping process exploits strong ground state transitions to feed metastable electronic states, causing relative changes in the transport properties, which can be measured using drift time spectrometers [2]. However, in the perspective of an application of the LRC method in the field of superheavy elements, the question of how well optical lines are defined becomes important, because atomic levels are simply missing from conventional tables. In this context, theoretical models play a significant role in calculating the electronic structure and predicting energies. Additionally, calculations of the transport properties involving the interaction between metal ions and rare gas elements are very useful in assessing experimental parameters such as the required detector sensitivities and beamtimes [3,4].

High-accuracy theoretical predictions for the heaviest elements should be based on atomic calculations involving relativistic methods and the many-body theory. For spectra, these problems are often solved by using the Fock Space Coupled Cluster (FSCC) [5–7], configuration Interaction (CI) models based on multiconfigurational Dirac–Hartree–Fock (MCDHF) [8–11], a combination of CI and the many-body perturbation theory (CI+MBPT) [12,13], or multireference (MRCI) theory [14–16]. FSCC is one of the most powerful available approaches that provides very accurate results at a reasonable computational price, where applicable. The limitation of FSCC is in its formulation which, until recently, could only accommodate up to two holes or two electrons. Lately, this method has been extended to treat three valence electrons [17]. The use of MRCI techniques provides flexibility, allowing investigations of various configurations, and considerable effort has been invested in making the CI algorithm functional within realistic computational resources. The MRCI results for heavy and superheavy elements can be found in the literature [14–16,18–20].

In this work, calculations of the electronic structures and the properties of $Lu^+$ and $Lr^+$ ions using the MRCI model are reported. These results are compared with the experimental data [21,22] and earlier FSCC theoretical findings for the energy levels and earlier CI+MBPT data for the electronic transition rates [23]. These comparisons are used to evaluate the reliability of the calculated energy levels and to gain insight into the prospects of using this method to further study superheavy ions with multiple valence electrons (more than two) and also to evaluate the molecular systems of metal ions and rare gas elements.

## 2. Theoretical Method

The calculations were carried out using the 2019 release of the DIRAC code [24,25]. The electronic structure and wavefunctions were computed based on the four-component Dirac–Coulomb Hamiltonian to ensure full relativistic treatment of the superheavy elements. The nuclei were described within the finite-nucleus model in the form of a Gaussian charge distribution [26]. The Dyall basis set series of double- (cv2z), triple- (cv3z) and quadruple-zeta (cv4z) cardinal numbers for both the Lu and Lr elements [27,28] were used. All the properties were computed with these basis sets, thus allowing us to also extrapolate the energy levels at the complete basis set (CBS) limit. The small component wavefunctions were generated from the large component basis sets by strict kinetic balance [29]. Further augmentation of the basis sets with extra diffuse functions in an even-tempered manner would not significantly impact the results, as found in preliminary tests made with single- and double-augmented calculations.

We divided the theoretical procedure into three steps. The first step consisted of the atomic Dirac–Hartree–Fock (DHF) calculations that were conducted by using the average of configuration (AOC) method [30]. We considered the AOC-type calculation for consistency with an earlier study of $Rf^+$ ions [31]. We also note that based on the AOC electronic structure, we always obtained reliable energy levels and transitions in heavy metal ions and their molecular complexes [32–35]. The AOC method was used to distribute the two valence electrons of the $Lu^+$ and $Lr^+$ ions within 12 valence spinors of the $s$ and $d$ atomic characters. In other words, we used fractional occupation numbers (0.1667 = 2/12) for the merged Lu $6s$ and $5d$ orbitals as well as for the Lr $7s$ and $6d$ orbitals, allowing us to obtain a totally symmetrical wavefunction that was isomorphic with the configuration system under which the MRCI model (see below) was operated.

The second step consisted of the MRCI calculations that were conducted based on the AOC DHF wavefunctions. The MRCI calculations were performed by using the Kramers-restricted configuration interaction module in the DIRAC code [24]. Table 1 shows the theoretical scheme for the generalized active space (GAS) [15,18,19] that was defined in the model. In total, 34 electrons were activated that formed the basis of the valence $5d$, $6s$ and $6p$ spinors of Lu (and similarly, $6d$, $7s$ and $7p$ of Lr) and the semi-core $4d$, $5s$, $5p$ and $4f$ spinors of Lu (and similarly, $5d$, $6s$, $6p$ and $5f$ of Lr). No excitations were allowed in GAS 1 in order to reduce the computational demands, whereas single- and double-electron excitations were allowed in GAS 2 and GAS 3, respectively, to complete the CI expansion (see Table 1). Virtual spinors with energies below 30 atomic units were also added in the CI expansion. The numbers of the requested roots in the MRCI calculations were adjusted to contain all the multiplet manifolds of the Lu (and Lr) $6s^2$ ($7s^2$), $5d^1 6s^1$ ($6d^1 7s^1$) and $6s^1 6p^1$ ($7s^1 7p^1$) configurations. In order to correct the energy levels for the Breit (transverse photon interaction) and the lowest-order quantum electrodynamics (QED) contributions (vacuum polarization and the self-energy terms) [36,37], we used the GRASP program package [38], which is based on the Dirac–Coulomb–Breit Hamiltonian and the multiconfiguration Dirac–Hartree–Fock (MCDHF) model. Aside from that, in the GRASP program [38], the self-energy terms are treated within the Welton approach, where the screening coefficients are approximated by the ratio of the Dirac wavefunction density in a small region around the nucleus to the same density obtained for hydrogenoic orbitals [39]. The reference spaces for the MCDHF calculations were the $4f^{14}(5d6s6p)^2$ and $5f^{14}(6d7s7p)^2$ multiplet manifolds of $Lu^+$ and $Lr^+$, respectively. For $Lu^+$, the virtual space for the CI expansion consisted

of one extra spinor for each *l* quantum number from 0 to 4 (i.e., 7*s*7*p*6*d*5*f*5*g*). For Lr$^+$ on the other hand, the virtual space consisted of one extra spinor for each *l* quantum number from 0 to 2, together with two extra spinors for each *l* quantum number from 3 to 5 and the 6*h* function (i.e., 8*s*8*p*7*d*6*f*7*f*5*g*6*g*6*h*). The Lu core 5*s* and 5*p* and Lr 6*s* and 6*p* electrons were also correlated. For each energy level, the quantities $\Delta_B$ and $\Delta_{B+QED}$ were calculated, representing the differences in the MCDHF energy without and with the Breit contributions and the differences in the MCDHF energy without and with the Breit+QED contributions, respectively.

**Table 1.** Specification of the generalized active space (GAS) scheme used for the calculations for the Lu$^+$ and Lr$^+$ ions (see the text for details).

| GAS | Accumulated Electrons | | Number of Spinors | Characters [a] |
|---|---|---|---|---|
| | Min [b] | Max | | |
| 1 | $10 - x$ | 10 | 10 | $(n-2)d$ |
| 2 | $18 - y$ | 18 | 8 | $(n-1)s$, $(n-1)p$ |
| 3 | $32 - z$ | 32 | 14 | $(n-2)f$ |
| 4 | 32 | 34 | 18 | $ns$, $(n-1)d$, $np$ |
| 5 | 34 | 34 | (<30 au) [c] | Virtual |

[a] For Lu$^+$ and Lr$^+$, n = 6 and 7, respectively. [b] $x$, $y$ and $z$ are variables that control the electron excitation process attributed to the selective GAS. In the calculations, we defined the following: $x = 0$, $y = 2$ and $z = 1$, in line with the previous presentation [31]. [c] This includes all the virtual spinors up to an energy of 30 atomic units.

The third step consisted of the calculation of the spectroscopic properties based on the transition dipole moment between the levels. We used the relativistic transition moment operator within the MRCI method [40,41] to derive the oscillator strengths of the electronic transitions at the electric dipole ($E_1$) level. We considered the multiplet manifolds of the Lu 6*s*$^2$ and 5*d*$^1$6*s*$^1$ (Lr 7*s*$^2$ and 6*d*$^1$7*s*$^1$) configurations as the lower levels and the multiplet manifold of the Lu 6*p*$^1$6*s*$^1$ (Lr 7*p*$^1$7*s*$^1$) configurations as the upper ones. The calculations of the Einstein coefficients and branching ratios were also conducted by following the standard equations [42].

## 3. Results

Table 2 lists the energies of the ground and the low-lying excited states of the Lu$^+$ ions as obtained from the MRCI calculations. We used the natural orbital occupation numbers of the CI vectors to deduce the dominant electron configuration of each electronic state. The electronic states were predominantly the multiplet manifold of the 6*s*$^2$, 5*d*$^1$6*s*$^1$ and 6*s*$^1$6*p*$^1$ configurations (see Table 2). Note that the multiplets that originated from configurations 5*d*$^2$ or 5*d*$^1$6*p*$^1$ were omitted for convenience because they were found to be higher in energy. For comparison, Table 2 also shows the reference energies that were taken from the literature (i.e., the experimental data collected within the framework of the National Institute of Standards and Technology (NIST) atomic spectra database [21] and previous calculations based on the FSCC model [23]).

The four columns that are depicted in the MRCI results section of Table 2 show the calculated energy levels for the three different basis sets and, subsequently, the energy extrapolated to the MRCI complete basis set limit ($E(\infty)$). To derive ($E(\infty)$), we used the polynomial ($n^{-3}$) complete basis set scheme [43] for the correlation energies with cardinal number $n = 3$ and 4 for triple and quadruple zeta, respectively. In Table 2, the calculated energy corrections for the Breit and Breit+QED contributions are also listed, together with the final values that add up the MRCI CBS limit and the energy corrections. For these final values (see Table 2), the numbers in brackets indicate the likely uncertainties due to the computational protocol in the least significant digits of the energy values. The uncertainties consist of the absolute value of the difference in energy between the data obtained with the triple-zeta and quadruple-zeta basis sets. Table 3 lists the calculated energies for the ground and low-lying excited states of Lr$^+$ ion as obtained from the MRCI calculations, together

with the energy corrections due to the Breit interactions and QED. Similar to Table 2, the final values also include the likely uncertainties due to the computational protocol. The energy spectrum of $Lr^+$ was very similar to that of the $Lu^+$ homolog. The ground and low-lying excited states belonged to the multiplet manifold of configurations $7s^2$, $6d^17s^1$ and $7s^17p^1$. For comparison, Table 3 also shows the earlier FSCC results [23].

**Table 2.** Calculated energies (in $cm^{-1}$) of the ground and the low-lying excited states of the $Lu^+$ ions obtained from the MRCI model using the double- (2), triple- (3) and quadruple-zeta (4) basis sets and the energy values derived at the complete basis set limit ($\infty$), together with the final energy values (Final) that take into consideration the energy corrections obtained for the Breit ($\Delta_B$) and QED ($\Delta_{B+QED}$) contributions, compared with the experimental data (Exp.) and the FSCC results.

| Levels | | | MRCI | | | | Corrections | | Final | Reference | |
| --- | --- | --- | --- | --- | --- | --- | --- | --- | --- | --- | --- |
| Config. | State | $J$ | (2) | (3) | (4) | ($\infty$) | $\Delta_B$ | $\Delta_{B+QED}$ | | Exp. [a] | FSCC [b] |
| $6s^2$ | $^1S$ | 0 | 0 | 0 | 0 | 0 | | | 0 | 0 | 0 |
| $5d^16s^1$ | $^3D$ | 1 | 12,227 | 12,213 | 12,172 | 12,145 | 92 | −104 | 12,041 (41) | 11,796 | 12,354 |
| | | 2 | 12,698 | 12,669 | 12,626 | 12,598 | 91 | −88 | 12,510 (43) | 12,435 | 12,985 |
| | | 3 | 13,946 | 13,907 | 13,866 | 13,838 | 88 | −24 | 13,814 (41) | 14,199 | 14,702 |
| | $^1D$ | 2 | 16,817 | 16,656 | 16,583 | 16,535 | 98 | −44 | 16,491 (73) | 17,333 | 17,892 |
| $6s^16p^1$ | $^3P$ | 0 | 27,712 | 28,004 | 28,462 | 28,752 | 63 | −88 | 28,664 (456) | 27,264 | 27,091 |
| | | 1 | 28,886 | 29,208 | 29,646 | 29,923 | 64 | −77 | 29,846 (438) | 28,503 | 28,440 |
| | | 2 | 32,650 | 33,127 | 33,599 | 33,899 | 60 | −36 | 33,863 (472) | 32,453 | 32,294 |
| | $^1P$ | 1 | 38,071 | 38,402 | 38,453 | 38,484 | 101 | −51 | 38,433 (51) | 38,223 | 38,464 |

[a] Taken from [21,22]. [b] Taken from [23].

**Table 3.** Calculated energies (in $cm^{-1}$) of the ground and the low-lying excited states of the $Lr^+$ ion obtained from the MRCI model using the double- (2), triple (3) and quadruple-zeta (4) basis sets, and the energy values derived at the complete basis set limit ($\infty$), together with the final energy values (Final) that take into consideration the energy corrections obtained for the Breit ($\Delta_B$) and QED ($\Delta_{B+QED}$) contributions, compared with the FSCC results.

| Levels | | | MRCI | | | | Corrections | | Final | Reference |
| --- | --- | --- | --- | --- | --- | --- | --- | --- | --- | --- |
| Config. | State | $J$ | (2) | (3) | (4) | ($\infty$) | $\Delta_B$ | $\Delta_{B+QED}$ | | FSCC [a] |
| $7s^2$ | $^1S$ | 0 | 0 | 0 | 0 | 0 | | | 0 | 0 |
| $6d^17s^1$ | $^3D$ | 1 | 21,796 | 21,768 | 21,696 | 21,649 | 219 | −86 | 21,563 (72) | 20,265 |
| | | 2 | 22,494 | 22,459 | 22,375 | 22,320 | 218 | −61 | 22,259 (84) | 21,623 |
| | | 3 | 24,761 | 24,723 | 24,633 | 24,574 | 211 | 56 | 24,630 (90) | 26,210 |
| | $^1D$ | 2 | 28,883 | 28,721 | 28,570 | 28,472 | 230 | 32 | 28,504 (151) | 31,200 |
| $7s^17p^1$ | $^3P$ | 0 | 29,825 | 30,072 | 31,006 | 31,600 | 144 | −81 | 31,519 (934) | 29,487 |
| | | 1 | 32,114 | 32,360 | 33,222 | 33,770 | 150 | −60 | 33,710 (862) | 31,610 |
| | | 2 | 43,428 | 43,809 | 44,783 | 45,402 | 152 | 49 | 45,451 (974) | 43,513 |
| | $^1P$ | 1 | 47,908 | 48,135 | 48,794 | 49,212 | 205 | 33 | 49,245 (659) | 47,819 |

[a] Taken from [23].

Table 4 lists the spectroscopic properties obtained for the $Lu^+$ ions. The upper energy electronic states that belong to the configuration $6s^16p^1$ decayed via the electric dipole $E_1$ mechanism to the lower energy states from configurations $5d^16s^1$ and $6s^2$. To obtain the oscillator strengths, the Einstein coefficients and the branching ratios, we used the transition dipole moments obtained with the MRCI model [24,25], while we considered the extrapolated energy for the complete basis set limit in Table 2 for the $\Delta E$ between the upper and the lower energy levels. In Table 4, we also report the available experimental data for the $Lu^+$ ions for comparison [21,22].

Finally, Table 5 lists the spectroscopic properties obtained for the $Lr^+$ ions, alongside theoretical predictions that were taken from the literature (CI+MBPT model) [23]. Similar to the $Lu^+$ homolog, we show the transition rates for the multiplet manifolds of configuration $7s^17p^1$ that decay via the electric dipole $E_1$ mechanism to the multiplet manifolds of the configurations $6d^17s^1$ and $7s^2$. We observed that the calculated electronic transition rates of $Lr^+$ were slightly larger than those calculated for the $Lu^+$ homolog, and they were in good agreement with the earlier theoretical data taken from the literature [23].

**Table 4.** Calculated Einstein coefficients $A_{E1}$ (in 1/s) and branching ratios $\beta$ for the electric dipole's allowed transitions in $Lu^+$, obtained from the MRCI transition dipole moment matrix and the $\Delta E$ (in $cm^{-1}$) from the complete basis set limit, compared with the reference experimental values.

| Levels | | MRCI | | | Reference [a] |
|---|---|---|---|---|---|
| Upper | Lower | $\Delta E$ | $A_{E1}$ | $\beta$ | $A_{E1}$(NIST) |
| $^3P_1$ ($6s^16p^1$) | $^1S_0$ ($6s^2$) | 29,924 | $6.10 \times 10^6$ | 0.08 | $1.25 \times 10^7$ |
| $^1P_1$ ($6s^16p^1$) | | 38,474 | $3.74 \times 10^8$ | 0.90 | $4.53 \times 10^8$ |
| $^3P_0$ ($6s^16p^1$) | $^3D_1$ ($5d^16s^1$) | 16,609 | $4.38 \times 10^7$ | 1.00 | |
| $^3P_1$ ($6s^16p^1$) | | 17,779 | $1.39 \times 10^7$ | 0.19 | |
| $^3P_2$ ($6s^16p^1$) | | 21,750 | $1.14 \times 10^6$ | <0.01 | |
| $^1P_1$ ($6s^16p^1$) | | 26,329 | $4.00 \times 10^5$ | <0.01 | |
| $^3P_1$ ($6s^16p^1$) | $^3D_2$ ($5d^16s^1$) | 17,326 | $5.36 \times 10^7$ | 0.72 | $9.90 \times 10^6$ |
| $^3P_2$ ($6s^16p^1$) | | 21,297 | $2.06 \times 10^7$ | 0.16 | |
| $^1P_1$ ($6s^16p^1$) | | 25,876 | $3.48 \times 10^7$ | 0.08 | |
| $^3P_2$ ($6s^16p^1$) | $^3D_3$ ($5d^16s^1$) | 20,058 | $1.09 \times 10^8$ | 0.82 | |
| $^3P_1$ ($6s^16p^1$) | $^1D_2$ ($5d^16s^1$) | 13,385 | $1.04 \times 10^6$ | 0.01 | |
| $^3P_2$ ($6s^16p^1$) | | 17,356 | $1.82 \times 10^6$ | 0.01 | |
| $^1P_1$ ($6s^16p^1$) | | 21,935 | $7.95 \times 10^6$ | 0.01 | |

[a] Taken from [21,22].

**Table 5.** Calculated Einstein coefficients $A_{E1}$ (in 1/s) and branching ratios $\beta$ for the electric dipole's allowed transitions in $Lr^+$, obtained from the MRCI transition dipole moment matrix and the $\Delta E$ (in $cm^{-1}$) from the complete basis set limit, compared with the reference theoretical data.

| Levels | | MRCI | | | Reference [a] |
|---|---|---|---|---|---|
| Upper | Lower | $\Delta E$ | $A_{E1}$ | $\beta$ | $A_{E1}$(CI+MBPT) |
| $^3P_1$ ($7s^17p^1$) | $^1S_0$ ($7s^2$) | 33,783 | $2.97 \times 10^7$ | 0.49 | $6.36 \times 10^7$ |
| $^1P_1$ ($7s^17p^1$) | | 49,221 | $7.93 \times 10^8$ | 0.87 | $8.34 \times 10^8$ |
| $^3P_0$ ($7s^17p^1$) | $^3D_1$ ($6d^17s^1$) | 9966 | $1.54 \times 10^7$ | 1.00 | $5.44 \times 10^6$ |
| $^3P_1$ ($7s^17p^1$) | | 12,134 | $6.91 \times 10^6$ | 0.11 | $2.42 \times 10^6$ |
| $^3P_2$ ($7s^17p^1$) | | 23,764 | $2.44 \times 10^6$ | <0.01 | $9.41 \times 10^5$ |
| $^1P_1$ ($7s^17p^1$) | | 27,572 | $1.07 \times 10^6$ | <0.01 | $1.36 \times 10^6$ |
| $^3P_1$ ($7s^17p^1$) | $^3D_2$ ($6d^17s^1$) | 11,463 | $2.38 \times 10^7$ | 0.39 | $4.66 \times 10^6$ |
| $^3P_2$ ($7s^17p^1$) | | 23,093 | $4.03 \times 10^7$ | 0.17 | $9.70 \times 10^6$ |
| $^1P_1$ ($7s^17p^1$) | | 26,901 | $4.98 \times 10^7$ | 0.06 | $1.63 \times 10^7$ |
| $^3P_2$ ($7s^17p^1$) | $^3D_3$ ($6d^17s^1$) | 20,839 | $1.93 \times 10^8$ | 0.81 | $3.43 \times 10^7$ |
| $^3P_1$ ($7s^17p^1$) | $^1D_2$ ($6d^17s^1$) | 5307 | $2.51 \times 10^4$ | <0.01 | |
| $^3P_2$ ($7s^17p^1$) | | 16,937 | $2.68 \times 10^6$ | 0.01 | $3.19 \times 10^5$ |
| $^1P_1$ ($7s^17p^1$) | | 20,745 | $6.60 \times 10^7$ | 0.07 | $1.68 \times 10^7$ |

[a] Theoretical values obtained by using CI plus many-body perturbation theory (MBPT) in [23].

## 4. Discussion

The $Lu^+$ and $Lr^+$ ions exhibited the same closed shell ground states and very similar energy spectra. The low-lying excited states of both the $Lu^+$ and $Lr^+$ ions belonged to the configurations $5d^16s^1$ and $6d^17s^1$, respectively. The multiplet manifolds of the Lu $6s^16p^1$ and

Lr $7s^1 7p^1$ configurations were higher in energy for both ions. The energy splitting in Lr$^+$ was larger than Lu$^+$ because of the larger spin–orbit interaction expected for the heavier element. We found that the Breit and QED corrections were relatively small, being to the order of 100 and 200 cm$^{-1}$ for Lu$^+$ and Lr$^+$, respectively, with energy values comparable to the Breit and QED effects calculated for analogous elements [23,31,44].

The MRCI energies of the Lu$^+$ ions were in good agreement with the experimental data [21,22] and previous theoretical findings [23] (see Table 2). The relative errors of most of the tabulated energy levels with respect to the experimental values were less than 5%. For the $5d^1 6s^1$ configuration, the term with $J = 2$ ($^1$D) had the highest error (4.6%), making the term more susceptible to the interaction with the higher energy levels of the $5d^2$ configuration. For configuration $6s^1 6p^1$, the term with $J = 0$ ($^3$P) had the highest uncertainty (5.5%), which might also be due to mixing with higher energy-excited electronic states.

The MRCI energies of the Lr$^+$ ions were also in good agreement with the previous theoretical findings [23] (see Table 3), with slightly larger deviations than for Lu$^+$ within the range of 2.9 % to 8.7 %. For the $6d^1 7s^1$ configuration, the term with $J = 2$ ($^1$D) had the largest deviation (8.7%), a level which is more susceptible to interaction with the $6d^2$ configuration which, in its multiplet manifold, possesses a term with the same symmetry ($6d^2 \longrightarrow \,^1$S + $^3$P + $^1$D + $^3$F + $^1$G). For the $7s^1 7p^1$ configuration, the terms with $J = 0$ and 1 ($^3$P) had the highest deviations (7.2 % and 6.9 %, respectively).

We note that the discrepancies from the reference values were larger for the odd parity states (Lu $6s^1 6p^1$ and Lr $7s^1 7p^1$) than those for the even parity states (Lu $5d^1 6s^1$ and Lr $6d^1 7s^1$). These might result from the choice of the reference spinors for the MRCI calculation, since the Lu $6p$ (as well as Lr $7p$) spinors were left outside of the AOC occupation scheme. A possible way to improve the odd parity energy levels would be to build another AOC occupation scheme by changing the occupation number to two electrons in the Lu $6s$ and $6p$ (as well as Lr $7s\ 7p$) and therefore run the MRCI calculation of the even and odd parity energy levels individually. The calculated Einstein coefficients for the Lu$^+$ inter-configurational $6s^2 \longrightarrow 6s^1 6p^1$ transitions were in good agreement with the experimental data [21,22], where three electric dipole transitions were reported. The strongest transition corresponded to the $^1$S$_0$ ($6s^2$) $\longrightarrow \,^1$P$_1$ ($6s^1 6p^1$), in line with the experimental data [21,22], but we noted the slight overestimation of the MRCI results (see Table 4). The calculated Einstein coefficients for the Lr$^+$ inter-configurational $7s^2 \longrightarrow 7s^1 7p^1$ transitions were also in agreement with the previously reported CI+MBPT values [23]. The strongest transition corresponded to the $^1$S$_0$ ($7s^2$) $\longrightarrow \,^1$P$_1$ ($7s^1 7p^1$), as was previously predicted [23]. The second strongest transitions corresponded to the $^3$D$_3$ ($6d^1 7s^1$) $\longrightarrow \,^3$P$_2$ ($7s^1 7p^1$) according to our MRCI calculation, unlike the $^1$S$_0$ ($7s^2$) $\longrightarrow \,^3$P$_1$ ($7s^1 7p^1$) transition predicted in [23].

## 5. Conclusions

In this work, we calculated the electronic energy levels and spectroscopic properties of the Lr$^+$ ions and of the homolog Lu$^+$ ions. We used a multireference model and configuration interaction approach to obtain the electronic structure and to compute the transition probabilities. The theoretical results were compared with the experimental data for the Lu$^+$ ions and previous theoretical findings for the Lr$^+$ ions. For Lu$^+$, the results were remarkably very close to the experimental data, allowing us to translate the theoretical procedure to treat the heavier Lr$^+$ ions. For this, the calculated energy levels were also consistent later with the previous theoretical findings based on the Fock Space-coupled cluster method. We conclude that MRCI is a reliable theoretical model in computing energy levels for heavy and superheavy elements. MRCI is potentially of interest for systems with more than two valence electrons and also for the calculation of the interaction between the metal ions and rare gas atoms. The latter will be used to describe the transport properties of these ions in our next theoretical development.

Our results support the conclusions from previous theoretical work. (1) The energy spectrum of the Lr$^+$ ion was predicted to be similar to the one obtained for the Lu$^+$ homolog, and (2) both the Lr$^+$ and Lu$^+$ ions are good candidates for future Laser Resonance

Chromatography experiments. In fact, their energy spectra present a case for experiments based on a metastable electronic state that is too long-lived for spectroscopy experiments. A potential LRC route consists of pumping the ground state $^1$S$_0$ ($6s^2$ and $7s^2$) to the excited state $^3$P$_1$ ($6s^16p^1$ and $7s^17p^1$), which radiatively decays to the metastable $^3$D$_1$ ($5d^16s^1$ and $6d^17s^1$) state with a sizeable branching ratio.

**Author Contributions:** Conceptualization, H.R.; methodology, H.R. and A.B.; software, H.R. and A.B.; validation, H.R., A.B., M.B. and M.L.; formal analysis, H.R., A.B. and M.L.; investigation, H.R.; resources, A.B. and M.B.; data curation, H.R. and A.B.; writing—original draft preparation, H.R.; writing—review and editing, A.B., M.B. and M.L.; visualization, H.R.; supervision, A.B. and M.L.; project administration, M.B. and M.L.; funding acquisition, M.L. All authors have read and agreed to the published version of the manuscript.

**Funding:** This project received funding from the European Research Council (ERC) under the European Union's Horizon 2020 Research and Innovation Programme (Grant Agreement No. 819957).

**Institutional Review Board Statement:** Not applicable.

**Informed Consent Statement:** Not applicable.

**Data Availability Statement:** Data is contained within the article.

**Acknowledgments:** We gratefully acknowledge the high-performance computing (HPC) support, time and infrastructure from the Center for Information Technology of the University of Groningen (Peregrine), the Johannes Gutenberg University of Mainz (Mogon) and the HPC group of GSI. We are also indebted to the HPC-Europa3 program (host: Anastasia Borschevsky; guest: Harry Ramanantoanina) for HPC computer time (Netherlands) and for the funding of a short-term scientific mission in Groningen.

**Conflicts of Interest:** The authors declare no conflict of interest.

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
