# Peer review of "Electronic Structure of Lr+ (Z = 103) from Ab Initio Calculations"

_atoms, doi:10.3390/atoms10020048_

Round 1

Reviewer 1 Report

The author calculated the electronic spectrum, Einstein Coefficients using the MRCI method. Although the calculations of these elements have been reported, the calculation at the MRCI level is new.
I recommend publication of the manuscript after the authors consider the following comments aiming to improve the manuscript.

>Line 33-37 and 40-42
The motivation of this manuscript is not explicitly mentioned in the introduction. 
I agree with the advantage of the MRCI (e.g., flexibility), but for the target of your manuscript, the low-energy excited states of Lu+ and Lr+ atoms, the FSCC in the previous study works much better than MRCI. In principle, the authors do not have to run MRCI calculations for these systems. 
Although the authors mention the motivation in line 40-42, I propose to the authors that the motivation more clearly. Maybe the authors refer to line 180-181 to emphasize the systems with more than two valence electrons. 

> Line 63-66
Reference spinors employed by the authors are built with the AOC method, where two valence electrons are in 6s and 5d (7s and 6d) orbitals. It would work well for the calculations of the D states but would not be suitable for the P states. A better way for the P state is the change of the occupation number to two electrons in 6s6p (7s7p) and running the calculations individually. In fact, the discrepancies from the reference values are larger for P states than those for D states. The authors should mention this point in the manuscript.

>line 116
The authors employ the exponential (e−αn) Dunning-Feller scheme [40,41] for the DHF total energy. However, the DHF energies of the ground and low-lying excited states should be the same because the authors' reference orbitals are the same: they are obtained with the AOC where two electrons are distributed in 5d6s (6d7s) spinors. The DHF energies are canceled out between the electronic and ground state, and it does not affect the final values. Maybe the authors do not have to mention it. 

> Line 135
In this one-line paragraph, the authors do not explain Table 5 at all. The authors should explain what new insight the author obtained from Table 5.

Next, I point out some tiny points. 

> whole manuscript
The quadrup"o"le-zeta basis set should be quadr"u"ple-zeta basis set.

> Line 26
The authors write, "by using the Fock Space Coupled Cluster (FSCC) [5–7],r"
The author should remove this strange "r".

> line 47
The authors write, "The nuclei were described within the finite-nucleus model in the form of a Gaussian charge distribution [26]."
And
[26] Visscher, L.; Dyall, K.G. Fully relativistic ab initio calculations of the energies of chiral molecules including parity-violating weak interactions. Atom. Data Nucl. Data Tabl. 1997, 67, 207. doi:10.1103/PhysRevA.60.4439. 

The authors should replace the title and doi with the correct ones. The current ones are wrong.

> Line 96
The authors write, "We used the relativistic transition moment operator within the MRCI method [24,25]".

[24,25] are the references for the DIRAC program. I propose the following ones as the references for the transition dipole moment (TDM). I should note that the TDM (the property module) had not been implemented in the initial stage of the GASCI module in the DIRAC code. The additional citations would be better.

(1) Stefan R. Knecht. Parallel Relativistic Multiconfiguration Methods: New Powerful Tools for Heavy-Element Electronic-Structure Studies. PhD thesis, Mathematisch-Naturwissenschaftliche Fakulta ̈t, Heinrich-Heine-Universit ̈at Du ̈sseldorf, 2009. 
(2) M. Denis, M. S. Norby, H. J. A. Jensen, A. S. P. Gomes, M. K. Nayak, S. Knecht, T. Fleig, New J. Phys. 17 (2015). doi:10.1088/1367-2630/17/4/043005. 

Reviewer 2 Report

In this work, the author investigate cations Lr+ and Lu+ by means of relativistic GAS-CI to predict their feasibility for LRC spectroscopic experiments. The computational strategy is clearly outlined and well justified by previous results. The results are of high accuracy as compared to experiment (where available) and previous theoretical studies.

I have a few comments:

  1. LRC is defined as Laser Resonance Spectroscopy. More commonly, C is reserved for Chromatography in this context.
  2. In Tables 2-3, I suggest to show Breit and QED contributions separately and add a column with the final value (CBS+Breit+QED).
  3. The final results would benefit from realistic error estimation associated with the computational protocol.
  4. Information about the self-energy model used to calculate the QED contribution is missing.
  5. Was double zeta basis set used for the CBS extrapolation of the correlation energy? This is typically not recommended.
  6. In Conclusions, it is stated that the Lr+ theoretical reference is based on FSCC, however, in Table 5 (as well as earlier in the text) it is stated that the method is CI+MBPT. This should be corrected.
